# The Role of Arrestin-1 Middle Loop in Rhodopsin Binding

**DOI:** 10.3390/ijms232213887

**Published:** 2022-11-11

**Authors:** Sergey A. Vishnivetskiy, Elizabeth K. Huh, Preethi C. Karnam, Samantha Oviedo, Eugenia V. Gurevich, Vsevolod V. Gurevich

**Affiliations:** 1Department of Pharmacology, Vanderbilt University, Nashville, TN 37232, USA; 2Department of Chemistry, The University of Texas at San Antonio, San Antonio, TX 78249, USA

**Keywords:** arrestin, GPCR, mutagenesis, protein–protein interactions, receptor binding, selectivity

## Abstract

Arrestins preferentially bind active phosphorylated G protein-coupled receptors (GPCRs). The middle loop, highly conserved in all arrestin subtypes, is localized in the central crest on the GPCR-binding side. Upon receptor binding, it directly interacts with bound GPCR and demonstrates the largest movement of any arrestin element in the structures of the complexes. Comprehensive mutagenesis of the middle loop of rhodopsin-specific arrestin-1 suggests that it primarily serves as a suppressor of binding to non-preferred forms of the receptor. Several mutations in the middle loop increase the binding to unphosphorylated light-activated rhodopsin severalfold, which makes them candidates for improving enhanced phosphorylation-independent arrestins. The data also suggest that enhanced forms of arrestin do not bind GPCRs exactly like the wild-type protein. Thus, the structures of the arrestin-receptor complexes, in all of which different enhanced arrestin mutants and reengineered receptors were used, must be interpreted with caution.

## 1. Introduction

Protein–protein interactions mediate the majority of regulatory processes in the living cell. As a rule, a particular functional state of one partner and/or certain post-translation modifications in it make it a preferred target for the interacting protein. The regulation of signaling of G protein-coupled receptors (GPCRs) follows these principles. G proteins and most isoforms of GPCR kinases preferentially bind activated GPCRs, while largely “ignoring” inactive receptors. Arrestins bind the active phosphorylated forms of their cognate receptors with much greater affinity than other functional forms of the same GPCR. Visual arrestin-1 (we use systematic names of arrestin proteins, where the number after the dash indicates the order of cloning: arrestin-1 (historic names S-antigen, 48 kDa protein, visual or rod arrestin), arrestin-2 (β-arrestin or β-arrestin1), arrestin-3 (β-arrestin-2 or hTHY-ARRX), and arrestin-4 (cone or X-arrestin)) demonstrates much greater selectivity than non-visual subtypes (reviewed in [1]). The binding of arrestin-1 to light-activated phosphorylated rhodopsin (P-Rh*) is 10–20 times greater than the binding to inactive phosphorylated (P-Rh) or active unphosphorylated (Rh*) rhodopsin, and virtually no binding to inactive unphosphorylated form can be detected [2]. This selectivity was explained by the existence of two independent sensors in arrestin, the phosphate and active receptor sensors, which only P-Rh* can engage at the same time [3]. This model suggested that arrestin-1 acts as a molecular coincidence detector: simultaneous engagement of the two sensors by P-Rh* promotes arrestin-1 transition into a high-affinity receptor-binding state, which brings additional elements into contact with the receptor [3]. The key corollary of this model is a significant conformational change in arrestin upon receptor binding. This idea was confirmed by extensive biophysical studies of arrestin-1 [4,5,6] and non-visual arrestin-2 and -3 [7], as well as by the comparison of the structures of free [8,9,10,11,12,13] and receptor (or non-receptor activator) bound arrestins-1, -2, and -3 [14,15,16,17,18,19,20,21]. The model implies that a large proportion of interaction energy (which directly translates into affinity) of an arrestin with its cognate receptor is contributed not by the sensors but by other receptor-binding elements that are brought into contact with a GPCR only when both sensors are engaged. The middle loop (earlier termed 139 loop in arrestin-1 [6]) of arrestin-1 is one of these elements.

Arrestin residues involved in phosphate binding were extensively studied by mutagenesis [22,23,24,25,26,27]. The key residues in the phosphate sensor were identified in visual arrestin-1 [15,27,28] and non-visual subtypes [25,27,29,30,31]. The role of the “finger” loop in the central crest of the receptor-binding arrestin side, which engages the cavity between transmembrane helices of the activated receptor in the available structures of the arrestin-receptor complexes [14,15,16,17,18,20,21,32], as the activation sensor in arrestin was recently established [33]. A nearby middle loop (Figure 1) is not a part of either sensor. However, this loop of arrestin-1 and arrestin-2 also comes into direct contact with unphosphorylated elements of their cognate receptors in solved structures of the complexes (reviewed in [34]) (Figure 1). Moreover, this loop demonstrates the greatest movement upon receptor binding in arrestin-1 (Figure 1) [6,14], as well as both non-visual subtypes [7]. Here we tested the effect of mutations of every residue in the middle loop in bovine arrestin-1 (Figure 2) on its binding to rhodopsin in the context of wild type (WT) arrestin-1 with high selectivity for P-Rh*, as well as in the context of two different “enhanced” mutants, truncated arrestin-1-(1-378) (Tr) and arrestin-1-(Phe375Ala, Val376Ala, Phe377Ala) (3A), both of which demonstrate much higher than WT binding to non-preferred forms of rhodopsin, unphosphorylated light-activated Rh* and inactive phosphorylated P-Rh [35]. The significance of this comparison is two-fold. One, various enhanced mutants of arrestin-1 and -2 were used in the structures of their complexes with bound GPCRs [14,15,16,17,18,20,21,32]. It is necessary to determine to what extent this structural information reflects the binding of WT proteins. Two, enahnced arrestin-1-3A mutant was used to compensate for defects in rhodopsin phosphorylation [36,37]. Similar mutants of non-visual subtypes were suggested as a potential therapy for disease-causing gain-of-function mutations of other GPCRs [38]. Understanding the molecular mechanism of the binding of the enhanced mutants to phosphorylated and unphosphorylated GPCRs is important for this endeavor. The data identified two residues in the middle loop that appear to participate in binding to any form of rhodopsin, as well as those that increase arrestin-1 selectivity for P-Rh* by suppressing the binding to Rh*.

## 2. Results

We used a direct binding assay to evaluate the effects of mutations in arrestin-1 on its binding to rhodsopsin. Produced in cell-free translation, radiolabeled WT and mutant arrestin-1 were incubated with P-Rh* and Rh*. Then bound arrestin-1 eluting with large rhodopsin-containing discs prepared from cow eyes was separated on size-exclusion column from much smaller free arrestin-1, as described in Methods. This assay yields the amount of bound arrestin-1 in each condition directly, does not require any assumptions, and therefore was previously used to characterize arrestin-1 interactions with different functional forms of rhodopsin [27,33,39,40].

The arrestin-1-rhodopsin interface in the structure of the complex is extensive: numerous arrestin-1 residues interact with rhodopsin [7,14]. Therefore, it is remarkable how many point mutations in the middle loop (previously called 139 loop [6]) in WT arrestin-1 significantly changed P-Rh* binding (Figure 3). It is even more surprising that the majority of substitutions (12 out of 21 tested) increased the binding to the preferred arrestin-1 target, P-Rh*, while only four reduced it. In most cases, the same mutation increased the binding to Rh* much more than to P-Rh* (Figure 3). These data suggest the middle loop with WT sequence primarily serves as a “break”, ensuring high selectivity of arrestin-1 for P-Rh*: native residues suppress the interaction with Rh*, often even at the expense of a slight reduction in the P-Rh* binding (Figure 3). Two mutations that increased the P-Rh* binding more than others (by 27–28%), D138A and V139A, exemplify this trend: they increased the Rh* binding by 180–240% (Figure 3). Both involve alanine substitutions of much bulkier side chains, one negatively charged, the other hydrophobic. In this context, the two mutations that reduced the P-Rh* binding by 20–25%, Q133E and Q137E, clearly stand out, as these substitutions also reduced the Rh* binding (Figure 3). The data suggest that negative charges in these positions are unfavorable for the binding to any form of the receptor, as both detrimental mutations were replacements of uncharged glutamine with negatively charged glutamic acid. Some mutations that did not change or only marginally affected the P-Rh* binding increased the binding to Rh* several-fold: P134G, ΔG140, and G140P (Figure 3). All of these involve proline or glycine, i.e., the residues that do not fit any secondary structure and, therefore, likely determine the overall shape of the loop. Apparently, the function of both Pro134 and Gly140 in the WT protein is to ensure the selectivity of arrestin-1 for P-Rh* by suppressing its binding to Rh*. This is consistent with earlier findings, which suggested that maintaining selectivity for P-Rh* is the key function of the 139-loop in arrestin-1: the deletion of residues 136–140 and 136–141 greatly reduced selectivity, increasing the binding to non-preferred functional forms of rhodopsin, inactive P-Rh and Rh* [6]. Both deletions also reduced arrestin-1 stability, particularly 136–141 deletion [6]. In contrast to WT arrestin-1, where the C-terminus is firmly anchored to the body of the N-domain [4,6,8], in about half of the molecules of the 136–140 deletion mutant, the C-tail was found to be released [6], as in the rhodopsin-bound arrestin-1 [4]. Thus, with the possible exception of Gln133 and Gln137, which likely directly participate in the interaction, the key functional role of the middle loop is to maintain arrestin-1 selectivity for P-Rh* by suppressing the Rh* binding. This is an unexpected finding.

If the middle loop in arrestin-1 both suppresses the binding to non-preferred forms of rhodopsin, as the data suggest (Figure 3), and participates in the interaction, as the structural evidence indicates [14,15], the effects of the mutations on the WT background may not reveal the possible direct role of individual middle loop residues in rhodopsin binding: positive effect of relaxed selectivity might mask the negative effect of the substitution due to possible loss of receptor interactions. Therefore, we introduced the same middle loop mutations on enhanced backgrounds, where the selectivity was already reduced by mutations outside of the middle loop. In this context, the data were expected to primarily reveal the direct effects of substitutions on arrestin-1 interactions with rhodopsin. To avoid mutant bias, two different enhanced versions were used: arrestin-1-(1-378) (Tr) and arrestin-1-(F375A, V376A, F377A) (3A). Both of these mutants demonstrate high binding to Rh* [35], which is very low in the case of WT (Figure 3).

The Tr background was successfully used recently to quantify the effects of various mutations on the arrestin-1 binding to unphosphorylated light-activated rhodopsin, which in the case of WT protein is too low to allow reliable comparisons [27,33]. For the sake of comprehensiveness, we determined the binding of all mutants on the Tr background to both P-Rh* and Rh*. Numerous mutations affected the P-Rh* binding on the Tr background: seven increased, while eight reduced it (Figure 4). This is in sharp contrast to the WT background, where twelve mutations enhanced and only four reduced the P-Rh* binding (Figure 3). This difference is consistent with the idea that the middle loop serves as a binding-reducing brake: partial release of the brake on the WT background would increase the binding, whereas, on the already relaxed background of enhanced mutant, loosening of the brake would not matter as much. Thus, the direct effects of mutations on the interaction with rhodopsin would come to the fore. On the Tr background, the most detrimental were D138R and S142A that reduced the P-Rh* binding by 24–30%, suggesting the involvement of the mutated residues in the arrestin-1 interaction with P-Rh*. The deletion of Pro136, as well as P134A and K141A mutations, increased the binding by 21–30%, suggesting that the native residues in these positions suppress the binding even in the context of the relaxed Tr mutant. Thus, the data on Tr background do not allow us to judge whether these residues play a direct role in rhodopsin interaction. The Rh* binding on the Tr background was also sensitive to middle loop mutations. D138A, D138R, and K141E decreased it by 21–41%, while Q133E was even more detrimental, decreasing the Rh* binding by more than 80% (Figure 4). These effects are consistent with the direct involvement of the mutated residues in the interaction of the Tr form of arrestin-1 with rhodopsin. Several mutations increased the Tr binding to Rh* by 20–35%: P134G, P134A, and K141A (Figure 4), indicating that proline in position 134 and lysine-141 act as the Rh* binding suppressors even on the background of the conformationally relaxed Tr mutant. Several mutations changed the P-Rh* and Rh* binding in the same direction: P134G, P134A, and K141A increased, whereas Q137E, D138A, D138R, and S142A decreased both. The most parsimonious interpretation of binding decreases is the direct participation of native residues in the interaction. Other mutations differentially affected the binding to these two forms of rhodopsin: the deletion of Pro136 increased only the P-Rh* binding, whereas the deletion of Gly140 and its substitution by proline decreased only the binding to P-Rh*. Differential effects of the same mutation on the binding to P-Rh* and Rh* likely suggest that the Tr mutant of arrestin-1 does not engage these two forms of rhodopsin in exactly the same manner. As both mutations were deletions of residues breaking the secondary structure of the middle loop, these results suggest that the WT shape of the loop is more important for the binding to the preferred arrestin-1 target, P-Rh*. K141E only slightly reduced the P-Rh* binding while suppressing the Rh* binding by more than 40% (Figure 4). The structural interpretation of this difference is not obvious. A lysine in this position was strictly conserved in all arrestin proteins for hundreds of millions of years of evolution [41], even though Lys- > Glu change would involve just one base substitution (AAG- > GAG or AAA- > GAA). This conservation suggests that the lysine in this position is functionally important. As a less drastic K141A mutation increased the binding to both forms of rhodopsin, it appears that this lysine acts as a strong binding suppressor.

The 3A mutant of arrestin-1 is also enhanced, demonstrating much greater binding to Rh* than WT [35]. In fact, a homologous 3A version of mouse arrestin-1 was shown in vivo to partially compensate for the lack of rhodopsin phosphorylation [36,37]. The 3A mutation acts via a mechanism that is similar but not identical to that operating in the Tr mutant. The C-tail, which is deleted in the Tr mutant, is present but detached from the N-domain in the 3A: the 3A mutation destabilizes its interactions with β-strand I and α-helix [8], mimicking the release of the C-tail observed upon rhodopsin binding [4,5,42]. The binding of mutants on the 3A background to both P-Rh* and Rh* was tested (Figure 5). As could be expected, the overall score was similar to that on the Tr background: seven mutations increased the binding to P-Rh*, while nine reduced it (Figure 5). However, the pattern of the effects on the 3A background was not identical to the pattern on the Tr background (compare Figure 4 and Figure 5). Q133E decreased the binding to P-Rh* on the 3A and WT backgrounds, but did not significantly affect it on the Tr background (Figure 3, Figure 4 and Figure 5). However, this mutation significantly reduced Rh* binding (by 50–80%) on all three backgrounds tested (Figure 3, Figure 4 and Figure 5). ΔP136 increased the P-Rh*, but not Rh*, binding on both Tr and 3A backgrounds while significantly increasing the Rh* binding of WT arrestin-1 (Figure 3, Figure 4 and Figure 5). This suggests that Pro136 suppresses the Rh* binding in WT protein but cannot play this role on conformationally relaxed backgrounds. D138R was detrimental for the binding to P-Rh* and Rh* on both enhanced backgrounds (Figure 4 and Figure 5), suggesting the direct participation of Asp138 in the interaction of these mutants with rhodopsin. However, the same mutation did not affect the P-Rh* binding of WT arrestin-1 (Figure 3). As D138R increased the Rh* binding of WT protein (Figure 3), native aspartic acid appears to act as a pure selectivity enhancer. ΔG140 increased P-Rh* and Rh* binding on WT and 3A backgrounds but did not have this effect on the Tr background (Figure 3, Figure 4 and Figure 5). K141A increased the P-Rh* and Rh* binding on both enhanced backgrounds but not in WT arrestin-1 (Figure 3, Figure 4 and Figure 5). K141E decreased the Rh* binding of Tr and 3A mutants by 41–44% (Figure 4 and Figure 5) but did not significantly affect it on a WT background (Figure 3). The effects of many other mutations (Q133A, ΔP134, P134A, A135L, Q137E, D138A, V139A, and ΔG140) also differ on the Tr and 3A backgrounds (Figure 4 and Figure 5). Collectively, the data suggest that WT and each of the two enhanced arrestin-1 mutants bind rhodopsin in a distinct manner: different residues appear to participate in the rhodopsin binding on these three backgrounds.

Finally, we tested whether the effects on the selectivity of binding-enhancing mutations in the middle loop are additive. To this end, we focused on mutations that increase Rh* binding and made the combination of P134G with ΔG140 in WT (cf. Figure 3), P134G + P136A on Tr background (cf. Figure 4), as well as P134G + ΔG140 and P134G + G140P on 3A background (cf. Figure 5). We reasoned that if these residues act independently of each other, the combinations would be additive or near-additive, but if the middle loop suppresses arrestin-1 binding to unpreferred forms of rhodopsin acting as a single unit, the simultaneous introduction of two mutations yielding similar phenotypes will not result in additive (increased compared to single mutations) effect. For the sake of comprehensiveness, we tested the effects of double mutations on P-Rh* and Rh* binding on all three backgrounds (Figure 6). We found that the effects were not additive in any case. On the WT background, the combination of two point mutations, P134G and ΔG140, both of which increase the binding to P-Rh* and Rh*, yields the same effect as the weakest of the two mutations, P134G (Figure 6). On the 3A background, double mutations P134G + G140P and P134G + ΔG140 reduced the P-Rh* binding and returned the Rh* binding back to the level of 3A alone, suggesting that in these combinations, individual mutations essentially cancel each other (Figure 6). On the Tr background, where individual P134G and P136A increase the binding to both P-Rh* and Rh*, the effect of P134G + P136A was not greater than that of the stronger P134G mutation (Figure 6). These data suggest that the middle loop acts as a whole as the selectivity suppressor. Apparently, its native shape (significantly perturbed by mutations affecting prolines and glycines) is an important requirement for its action, so simultaneous perturbations in two places do not enhance the effect. In some cases (e.g., in 3A mutant), the two perturbations cancel each other.

## 3. Discussion

A widely accepted model of arrestin binding to a GPCR explains arrestin selectivity for the active phosphorylated form of the receptor by the existence of two sensor sites, the phosphorylation and activation sensors, that must be engaged simultaneously to trigger arrestin transition into high-affinity receptor-binding state, which brings additional elements into contact with the receptor [1]. Our data suggest that this model needs to be modified, as the middle loop appears to function as a selectivity enhancer, suppressing the binding to non-preferred forms of rhodopsin, i.e., acting in addition to a simple coincidence mechanism proposed earlier.

The middle loop demonstrates the largest detected movement upon arrestin-1 [6] and arrestin-2 and -3 [7] binding to rhodopsin (Figure 1). It is also found in contact with the receptor in the structures of rhodopsin-bound arrestin-1 [14,15] (Figure 1) and receptor-bound arrestin-2 [16,17,18,20,21,32]. Thus, in addition to enhancing selectivity, the middle loop appears to directly contribute to the arrestin binding to the receptor. This loop is 11 residues long (Leu132 to Ser142 in bovine arrestin-1, Figure 2 and Figure 7). It is identical in bovine and mouse arrestin-1, and only one residue in it differs in the human protein [41]. It is also highly conserved in non-visual arrestin subtypes (Figure 7B). This loop contains hydrophobic and hydrophilic residues, including charged ones, as well as three residues that usually break secondary structure, Pro134, Pro136, and Gly140. To probe their role in rhodopsin binding, we substituted every residue with alanine, added a charge without affecting geometry (Gln- > Glu), introduced charge reversals (Asp- > Arg, Lys- > Glu), deleted Pro134 and Gly140, replaced these residues with Ala, as well as changed Pro into Gly and vice versa (Figure 2).

The most important finding is that the main function of the middle loop is to suppress arrestin-1 binding to Rh*, a non-preferred form of rhodopsin, thereby enhancing its selectivity for P-Rh*. This is novel and conceptually important, as this finding calls for a revision of the “coincidence detector” model of the arrestin-receptor interaction proposed earlier [1] and widely accepted in the field. A function like this cannot be revealed by structural work. The structures of basal [8,12,13] and receptor-bound [14,15] arrestin-1 reveal the starting and the final points of the process but do not show the process of protein–protein interaction itself, which likely includes multiple steps. Functional assays show how the changes in the interacting proteins affect the probability of the process reaching completion, i.e., resulting in actual high-affinity binding. These data uncover the existence of checkpoints that the binding process must pass through on the way to its completion. Molecular modeling integrating this information with structural data on the start and end point might suggest a plausible mechanism, which then needs to be tested by mutagenesis and functional assays. Our data revealed critical players important for selectivity enhancement: Pro134 and Pro136. As their mutations mostly increase the binding (Figure 3, Figure 4 and Figure 5), native Pro134 and Pro136 appear to suppress it (Figure 7A). The most plausible mechanism of their action is that these residues, which usually break secondary structure, confer a specific middle loop conformation in which it impedes arrestin-1 binding to non-preferred forms of rhodopsin, such as Rh*. Their substitutions appear to relieve this inhibition, thereby facilitating arrestin-1 binding to Rh*. This mechanism does not imply the direct participation of these residues in rhodopsin binding. Indeed, neither of these prolines is seen interacting with rhodopsin in the structure of the complex [14,15]. Quite a few other mutations on both WT (Figure 3) and enhanced (Figure 4 and Figure 5) backgrounds had a significant positive impact on the arrestin binding to rhodopsin. As virtually all of these mutations enhance the binding to Rh* significantly more than to P-Rh*, the most parsimonious explanation is that the native residues also increase arrestin-1 selectivity for P-Rh* over Rh*, in several cases at the expense of some reduction of the P-Rh* binding.

Data analysis also revealed two mutations, Q133E and S142A, that were nearly universally detrimental to rhodopsin binding (Figure 3, Figure 4 and Figure 5). This suggests the direct participation of the native residues in these two positions in the interaction of all forms of arrestin-1 with rhodopsin, either in the final complex or in the process of the two proteins adjusting to each other. Neither of these residues was implicated by structures of the arrestin-1-rhodopsin complex [14,15], likely because WT arrestin-1 has never been used for structural work. Curiously, alanine and glutamic acid are present in both WT non-visual subtypes where arrestin-1 carries serine and glutamine (Figure 7B). It would be interesting to test the effects of the substitution of these two residues with arrestin-1-specific serine and glutamine, respectively, on interactions of arrestin-2 and -3 with cognate non-visual GPCRs. If the mechanism of receptor binding is conserved, one would expect an enhancement. The substitutions of Asp138 in bovine arrestin-1 do not show uniform effects on the binding of the three forms of arrestin used, even though homologous residues in the arrestin-receptor complexes were found in contact with bound receptors: in mouse arrestin-1, homologous Asp139 interacts with rhodopsin, crosslinking with rhodopsin residues 148–150 [14], and in bovine arrestin-2 homologous Asp135 interacts with Arg65 in the bound muscarinic M2 receptor [17]. However, substitutions of Asp138 with alanine or even arginine with the opposite charge did not have a detrimental effect on the binding of WT arrestin to any form of rhodopsin (Figure 3). These data put the functional role of this residue in the arrestin-1 interaction with rhodopsin in doubt. As the effects of D138A and D138E mutations differ (Figure 3, Figure 4 and Figure 5), the data do not suggest that Asp138 participates in binding.

The sequence of this loop in arrestin-1, -2, -3, and -4 is remarkably conserved: in bovine proteins, seven out of eleven residues are identical in all four subtypes (Figure 7B). Conserved residues include two prolines and one glycine, i.e., three residues that usually break secondary structures. Three out of eleven residues constitute 27% of the total number. For comparison, in 404-residue full-length bovine arrestin-1, there are 42 prolines and glycines, or ~10% of the total [41]. A high proportion of secondary structure-breaking residues in the middle loop suggests that it is biologically important that it does not form any. Most likely due to these prolines and glycines, the middle loop assumes a particular shape, which appears to be functionally important. This is consistent with its unusually large movement upon receptor binding [6,14,15]. Homologous loop in non-visual subtypes arrestin-2 and -3 has even more secondary structure-breaking residues: two prolines and two glycines (36% of the total). The large movement of this loop upon receptor binding in arrestin-2 and -3 was also documented [7,16,17,18,32]. Notably, two mutations involving these residues, P134G and ΔP136, yielded an almost universal increase in binding (Figure 3, Figure 4 and Figure 5), suggesting that these prolines in WT arrestin-1 keep the binding in check, apparently for the sake of selectivity for P-Rh*.

The identification of binding-enhancing mutations has practical implications. The addition of mutations that further increase Rh* binding of enhanced mutants of arrestin-1 might improve in vivo compensational effect beyond that previously obtained with the 3A mutant [36,37]. As excessive signaling by non-visual GPCRs underlies a number of human disorders [38,43], the identification of mutations that increase receptor binding of non-visual arrestins also has practical implications. Mutations of homologous residues are good candidates for inclusion into potentially therapeutic enhanced versions of non-visual arrestins-2 and -3, where prolines in these positions in the middle loop are conserved [41] (Figure 7B). High binding to unphosphorylated GPCRs would make enhanced phosphorylation-independent arrestins effective tools for gene therapy of congenital disorders associated with defects in receptor phosphorylation.

Differential effects of numerous mutations on the WT, Tr, and 3A backgrounds (Figure 3, Figure 4 and Figure 5) suggest the involvement of distinct residues in these three forms of arrestin-1 in rhodopsin binding. In fact, out of 21 mutations tested, only two (Q133E and S124A) demonstrate the same pattern of effects on all three backgrounds, one (ΔG140) on the WT and 3A backgrounds, two (P134G and P134A) on the WT and Tr backgrounds, and two (ΔP136 and K141A) on 3A and Tr backgrounds. These data call for caution in the interpretation of available structures of the arrestin complexes with receptors, as mutationally enhanced arrestins, not the WT forms, were used in all studies: 3A arrestin-1 in the arrestin-1-rhodopsin complex [14,15], truncated arrestin-2 in complex with M2 muscarinic receptor [17], truncated cysteine-free arrestin-2 [32] or 3A mutant of arrestin-2 [16] in complex with the neurotensin receptor, enhanced R169E polar core mutant of arrestin-2 [30,31] in complex with the β1-adrenergic receptor [18], truncated arrestin-2 (1-382) in complex with the vasopressin V2 receptor [20], and doubly enhanced arrestin-2 (R169E mutant with deleted C-terminus) in complex with the 5HTB receptor [21]. It should also be noted that mutant receptors were also used in most structural studies: constitutively active rhodopsin containing E113Q and M257Y mutations [14,15], the β1-adrenergic receptor with deletions at both termini and in the third intracellular loop, in addition to nine point mutations and the replacement of it C-terminus with the phosphorylated C-terminus from the V2 vasopressin receptor [18], the non-glycosylatable vasopressin V2 receptor [20], and the 5HT2B receptor with deletions in the third loop and C-terminus plus mutations breaking the ionic lock that is believed to keep GPCRs in their inactive conformation [21]. The muscarinic M2 receptor used for structure determination was even farther than the others from WT [17]. Its large ( >150 residues) third intracellular loop was deleted, even though it contains determinants of its internalization [44] and carries all sites of agonist-induced phosphorylation [45], which are critical for arrestin binding [46]. While the WT M2 receptor has no phosphorylation sites in its short C-terminus, for structure determination, a phosphorylated V2 vasopressin receptor C-terminus was added by sortase [17]. The neurotensin [16,32] and vasopressin V2 [20] receptors used for determination of the structure of their complexes with arrestin-2 were the closest to the corresponding WT receptors. Curiously, in the vasopressin and neurotensin receptor complexes, the orientation of arrestin relative to the receptor was 38° and ~90° different from the other structures [16,20,32]. However, in case of the heavily mutagenized 5HT2B, the orientation of arrestin-2 relative to the bound receptor [21] was similar to that in case of the minimally modified V2 vasopressin receptor [20]. Given these caveats, alternative approaches are necessary to determine the involvement of individual WT arrestin and receptor residues in the interaction of these proteins. Site-directed mutagenesis is one of these alternatives. In this case, the proteins are changed as much or as little as needed to ask specific questions. Our data suggest that the arrestin-receptor interface likely differs in case of WT and mutant arrestin-1 binding. It also likely differs in case of cognate arrestin binding to WT and modified receptors. Thus, existing structures do not necessarily reveal true native interfaces. These might be identified more precisely by in-cell cross-linking of near-WT proteins with only single residue substitutions [47], although this method has its own caveats.

Our data indicate that the global shape of the WT middle loop is critically important for its function as a selectivity enhancer, as combinations of binding-enhancing mutations do not yield additive effects on any background (Figure 6). Structural studies available today did not yield this information. Our findings show that the coincidence detector mechanism proposed earlier [1] does not fully explain preferential arrestin binding to active phosphorylated receptors. The finding that an arrestin element serves as a selectivity enhancer calls for a refinement of the current model of the arrestin-receptor interaction. 

## 4. Materials and Methods

*Materials*. [γ-^32^P]ATP and [^14^C]leucine were purchased from Perkin-Elmer (Waltham, MA). Restriction endonucleases and T4 DNA ligase were from New England Biolabs (Ipswich, MA, USA). Amino acid-free rabbit reticulocyte lysate was from Ambion (Austin, TX, USA), and SP6 RNA polymerase was expressed and purified, as described [48]. DNA purification kits were from Zymo Research (Irvine, CA, USA). All other reagents were from Sigma-Aldrich (St. Louis, MO, USA).

*Mutagenesis and plasmid construction*. For in vitro transcription, bovine arrestin-1 was subcloned into a pGEM2 vector (Promega; Madison, WI, USA) with “idealized” 5-UTR [48] between Nco I and Hind III sites, as described [39]. Mutations were introduced by PCR between Pst I and Sal I unique restriction sites in a reengineered bovine arrestin-1 open reading frame [49] and subcloned into this construct. All mutations were confirmed by dideoxy sequencing (GenHunter Corporation, Nashville, TN, USA). Fragments containing mutant sequences were excised from sequence-confirmed wild-type (WT) clones with Pst I and Sal I and subcloned into constructs encoding Tr and 3A mutants.

In vitro *transcription, cell-free translation of uncapped mRNAs in rabbit reticulocyte lysate, calculation of specific activity of translated arrestin-1 mutants, as well as preparation of different functional forms of phosphorylated and unphosphorylated rhodopsin* were performed, as described recently [27,33,40,50].

*Direct binding assay* was performed, as described [39,40]. Briefly, 1 nM radiolabeled arrestin-1 (50 fmol) was incubated with 0.3 μg (~7.9 pmol) of indicated functional forms of rhodopsin in 50 μL of 50 mM Tris-HCL, pH 7.4, 100 mM potassium acetate, 1 mM EDTA, 1 mM DTT for 5 min at 37 °C under room light (both P-Rh* and Rh*). Samples were cooled on ice, whereupon bound and free arrestin-1 was separated at 4 °C by gel-filtration on 2-mL column of Sepharose CL-2B. Arrestin-1 eluting with rhodopsin-containing membranes was quantified by liquid scintillation counting (Tri-Carb Liquid Scintillation Analyzer, PerkinElmer, Waltham, MA, USA). Non-specific “binding” was determined in samples without rhodopsin and subtracted.

### Data Analysis and Statistics

Statistical significance (*p* < 0.05) was determined with one-way ANOVA (analysis of variance) with Dunnett’s multiple comparison test using GraphPad Prism software. In all cases, each middle loop mutant was compared to the corresponding parental protein. *p* values  <  0.05 were considered statistically significant and indicated as follows: * *p*  <  0.05; ** *p* < 0.01; *** *p*  <  0.001.

## Figures and Tables

**Figure 1 ijms-23-13887-f001:**
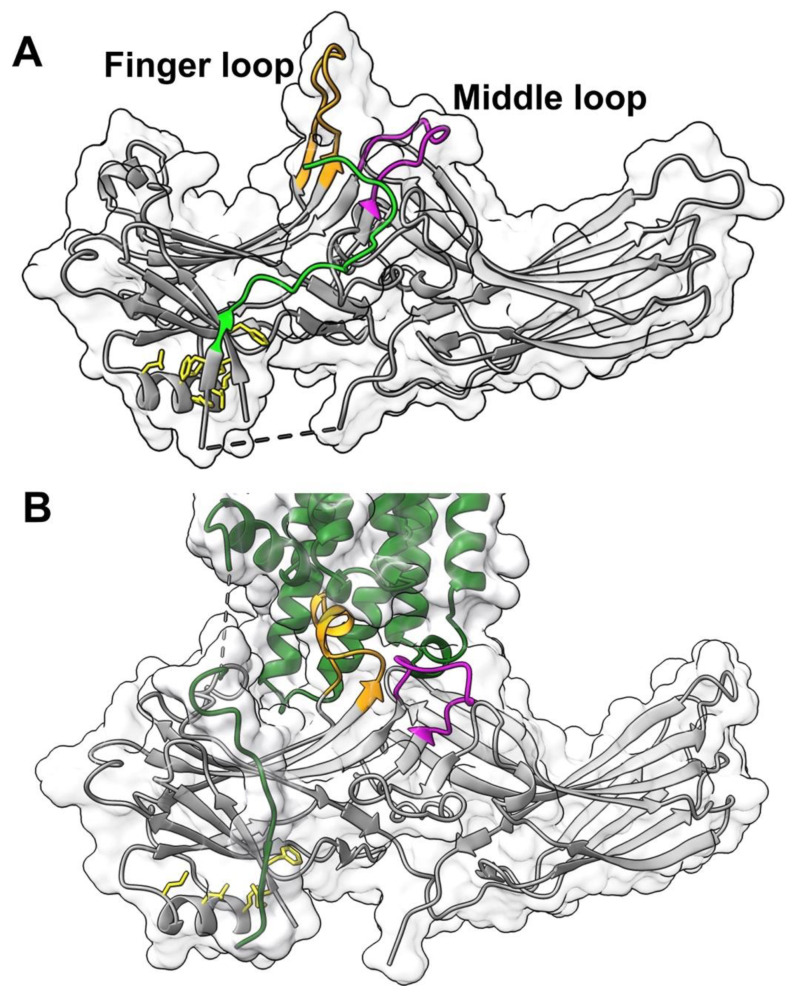
**The position of the middle loop in free and rhodopsin-bound arrestin-1.** (**A**) Arrestin-1 (molecule A in the crystal tetramer of bovine arrestin-1, PDB ID: 1cf1 [8]) with finger loop (residues 68–81) shown in orange and middle loop (residues 132–142) in magenta. Attached arrestin-1 C-terminus is shown in bright green. (**B**) The structure of the mouse arrestin-1 complex with rhodopsin (complex A, PDB ID: 5w0p [15]). Arrestin-1 (gray) in all panels and rhodopsin (dark green) in panel (**B**) are shown as flat ribbons with molecular surfaces indicated. The direction (N-to-C) of β-strands is shown by arrows. Side chains of bulky hydrophobic residues mediating the three-element interaction [8] are shown as yellow stick models. Images were created in DS ViewerPro 6.0 (Dassault Systèmes, San Diego, CA, USA).

**Figure 2 ijms-23-13887-f002:**
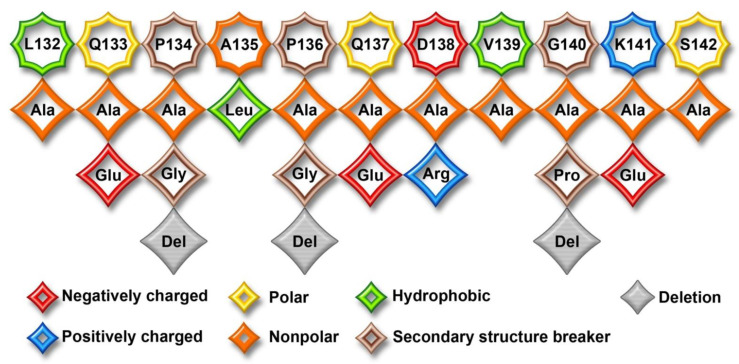
**Middle loop in arrestin-1 and the mutants.** Linear WT sequence of the middle loop of bovine arrestin-1 is shown on top, with residues as octagons. Substituting residues used in this study are shown as rhombi. The chemical nature of the original side chain and its replacements is shown by color, as indicated. Bulky hydrophobic residues were replaced with Ala and vice versa; charged residues were also replaced with Ala and a residue with the opposite charge; glutamines were replaced with Ala to remove the side chain and glutamic acid to introduce the charge without affecting the geometry; Gly and Pro that usually break the secondary structure were replaced by alanines (that are perfectly “happy” in any secondary structure), changed into each other, or deleted.

**Figure 3 ijms-23-13887-f003:**
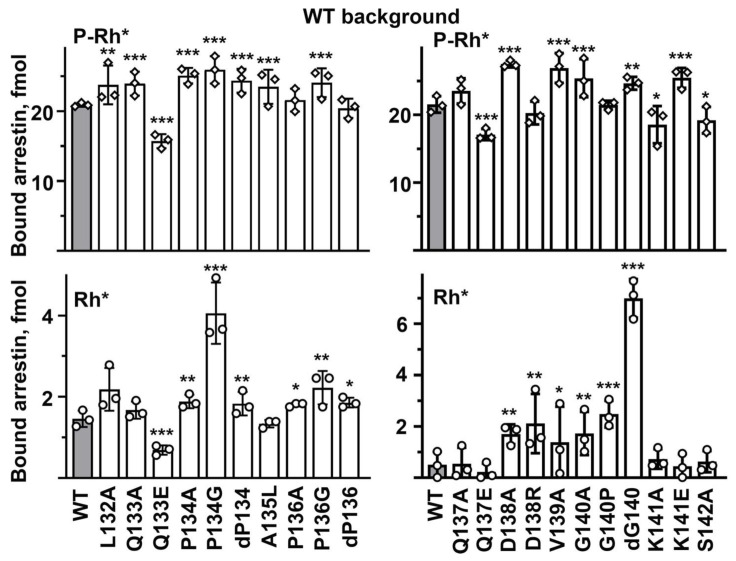
**The effect of middle loop mutations on WT background on arrestin-1 binding to rhodopsin.** The binding of indicated mutants of arrestin-1 to P-Rh* and Rh* was determined using radiolabeled arrestins, produced in cell-free translation, in the direct binding assay with purified phosphorylated or unphosphorylated light-activated bovine rhodopsin, as described in Methods. In the mutation names, “d” here and in the following figures stands for deletion, same as “Δ” in the text. Circles represent individual measurements performed in duplicate (*n* = 3). The binding to P-Rh* and Rh* was analyzed separately in each of the two groups. Statistical significance of the differences between WT (darker shaded bars) and the arrestin-1 mutants was determined by ANOVA followed by Dunnet post hoc test with correction for multiple comparisons and is indicated as follows: * *p*  <  0.05; ** *p* < 0.01; *** *p*  <  0.001 to WT.

**Figure 4 ijms-23-13887-f004:**
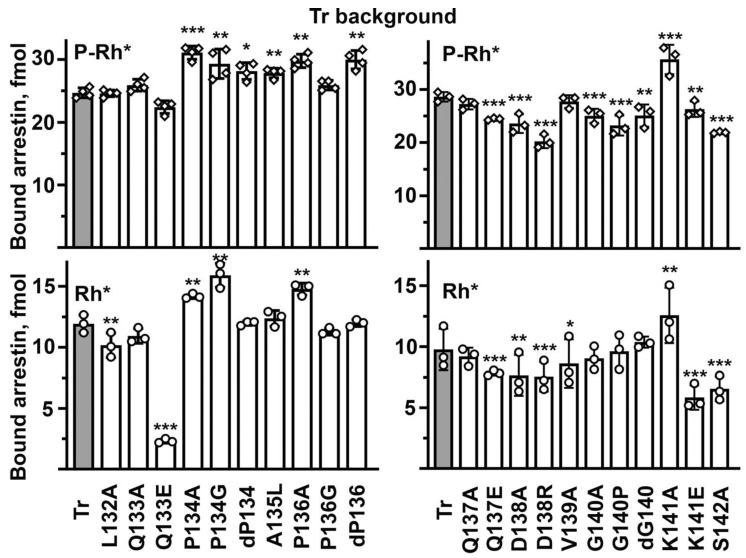
**The effect of middle loop mutations on Tr background on arrestin-1 binding to rhodopsin.** The binding of indicated mutants of arrestin-1 to P-Rh* and Rh* was determined using radiolabeled arrestins, produced in cell-free translation, in the direct binding assay with purified phosphorylated or unphosphorylated light-activated bovine rhodopsin, as described in Methods. Circles represent individual measurements performed in duplicate (*n* = 3). The binding to P-Rh* and Rh* was analyzed separately in each of the two groups. Statistical significance of the differences between the parental Tr (darker shaded bars) and the arrestin-1 mutants was determined by ANOVA and Dunnet post hoc test and is indicated as follows: * *p*  <  0.05; ** *p* < 0.01; *** *p*  <  0.001 to Tr.

**Figure 5 ijms-23-13887-f005:**
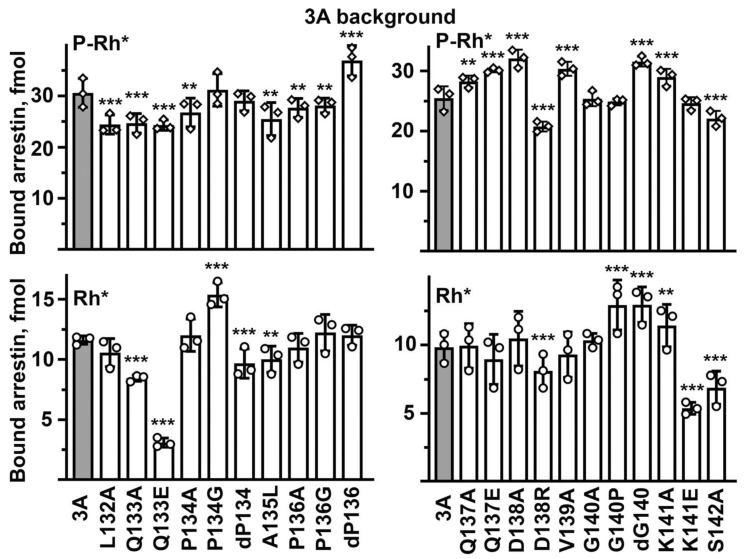
**The effect of middle loop mutations on 3A background on arrestin-1 binding to rhodopsin.** The binding of indicated mutants of arrestin-1 to P-Rh* and Rh* was determined using radiolabeled arrestins, produced in cell-free translation, in the direct binding assay with purified phosphorylated or unphosphorylated light-activated bovine rhodopsin, as described in Methods. Circles represent individual measurements performed in duplicate (*n* = 3). The binding to P-Rh* and Rh* was analyzed separately in each of the two groups. Statistical significance of the differences between the parental 3A (darker shaded bars) and the arrestin-1 mutants was determined by ANOVA and Dunnet post-test and is indicated as follows: ** *p* < 0.01; *** *p*  <  0.001 to 3A.

**Figure 6 ijms-23-13887-f006:**
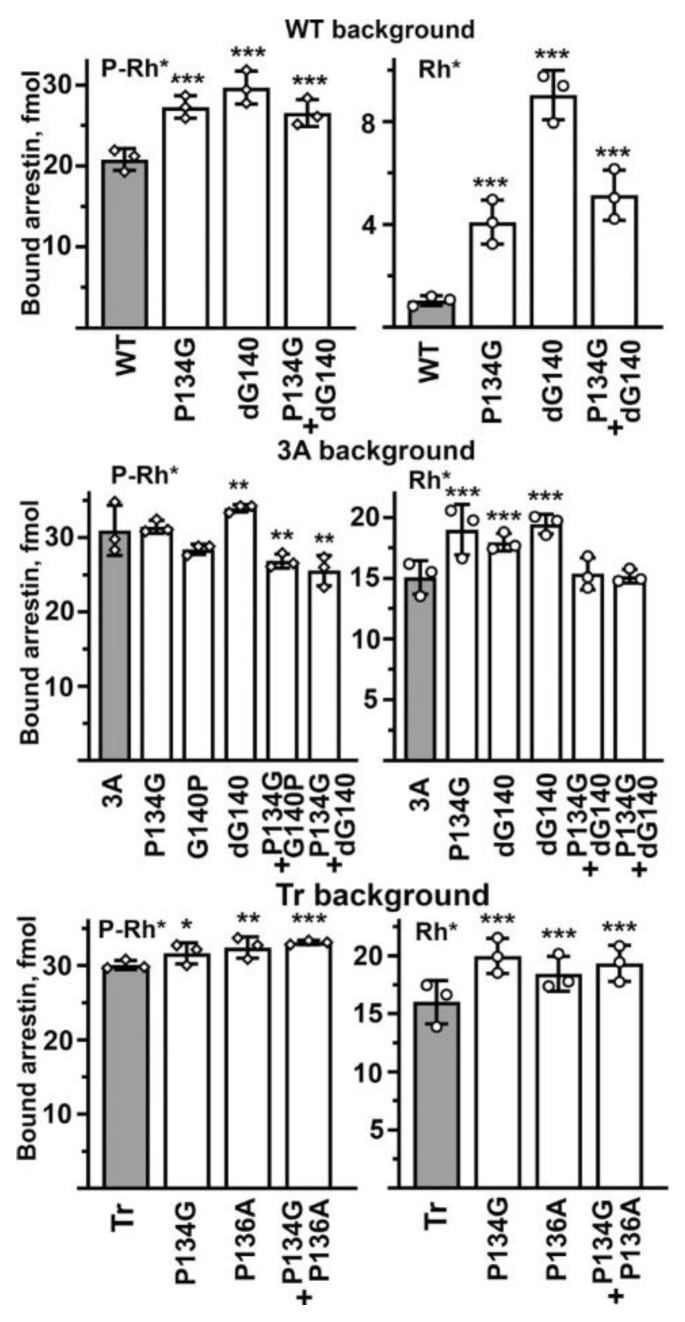
**Combinations of binding-enhancing mutations do not yield additive effects on any background.** The binding of indicated mutants of arrestin-1 to P-Rh* and Rh* was determined using radiolabeled arrestins, produced in cell-free translation, in the direct binding assay with purified phosphorylated or unphosphorylated light-activated bovine rhodopsin, as described in Methods. Circles represent individual measurements performed in duplicate (*n* = 3). The binding to P-Rh* and Rh* was analyzed separately in each group. Statistical significance of the differences between WT or background mutants (darker shaded bars) and the arrestin-1 middle loop mutants was determined by ANOVA and Dunnet post hoc test and is indicated as follows: * *p* < 0.05; ** *p* < 0.01; *** *p*  <  0.001 to WT, 3A, or Tr, respectively.

**Figure 7 ijms-23-13887-f007:**
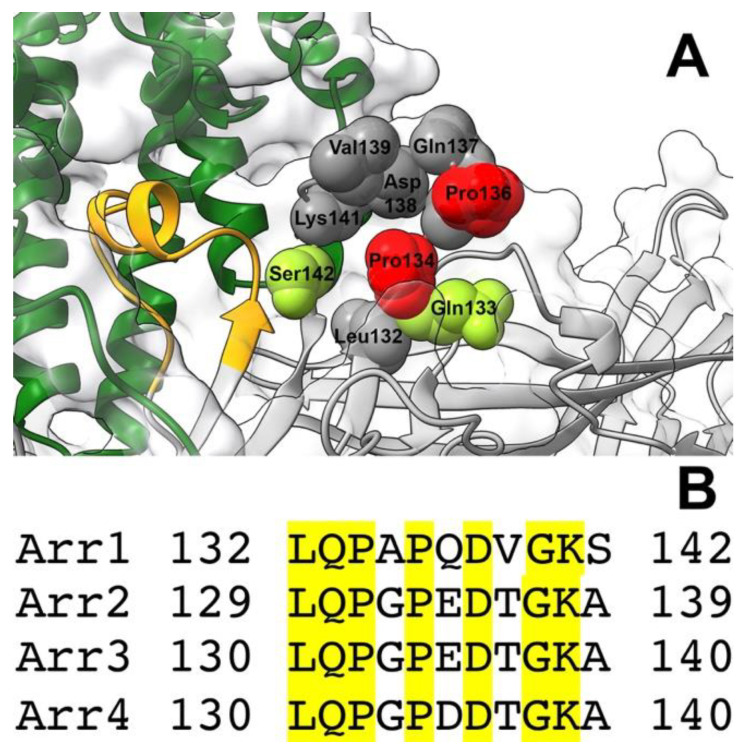
**Middle loop in arrestin family members.** (**A**) The structure of the middle loop in rhodopsin-bound arrestin-1 (complex A, PDB ID: 5w0p [7]). Arrestin (gray) and rhodopsin (dark green) are shown as ribbons with molecular surfaces indicated. The finger loop of arrestin-1 is shown in orange (arrow indicates N-to-C direction). Middle loop residues are shown as CPK models; two prolines in WT arrestin-1 that appear to suppress binding are shown in red; two residues important for binding (as the mutations of these residues reduce the binding on all backgrounds) are shown in light green; all other residues are shown in gray. Side chains are labeled with bovine arrestin-1 residue numbers (mouse arrestin-1 used in the structure has an identical middle loop, with the residue numbers N + 1, as compared to bovine protein). The image was created in DS ViewerPro 6.0 (Dassault Systèmes, San Diego, CA, USA). (**B**) The alignment of the sequences of middle loops of bovine arrestin-1, -2, -3, and -4. The numbers of the first and the last residue are indicated. Identical residues are shaded yellow.

## Data Availability

The data are presented in the manuscript. Raw binding data (in dpm or fmol) obtained in each experiment are available upon request.

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
