# Peer review of "The Role of Arrestin-1 Middle Loop in Rhodopsin Binding"

_ijms, 2022, doi:10.3390/ijms232213887_

Round 1

Reviewer 1 Report

The authors address an important question around the specificity of arrestin binding to activated GPCRs. A systematic mutational analysis has been performed to understand the contribution of individual amino acids to binding of phosphorylated GPCRs, with a particular focus on contribution of charge, structure and protein stability. The manuscript describes important findings that are interesting to the wider scientific community.

Author Response

Thank you!

Reviewer 2 Report

Review: ijms-1960528

In this work, the authors have reported that comprehensive mutagenesis within the middle loop of rhodopsin-specific arrestin-1 demonstrates that it serves as a suppressor of binding to non-preferred forms of the receptor. Their data also suggest that the enhanced forms of arrestin do not bind GPCRs comparable to the wild type.

This manuscript is important for understanding the role of arrestin-1 middle loop in rhodopsin binding. However, publication of this manuscript in its present form is not recommended.

To be considered further for publication, this work will need to be more organized and condensed in support of the claims made in the paper. Some specific points of concern are noted below:

1) The abstract need to be more organized. The following line should be rearranged:

“The data also suggest that enhanced forms of arrestin do not bind GPCRs exactly like the wild type protein, calling for caution in interpretation of structures of the arrestin-receptor complex, in all of which different enhanced arrestin mutants and reengineered receptors were used.”

2) On pg.11; paragraph 2, the authors mention that the mutations of residues Pro 134 and Pro136 increase the binding; this needs to be explained with more clarity. Some modeling experiments with ligand-receptor docking and scoring are necessary. The molecular interactions between arrestin and receptor will improve the quality of this paper.

3) The binding energy values of the WT arrestin - receptor and mutant arrestin- receptor complexes should be provided.

4) The uniqueness of the experimental methods used in this study needs to be explained with more clarity.

5) What are the applications of this work as for example the author briefly mention in the discussion section how some of these mutations can be used to design therapeutic protein structures.

6) Too many self citations.

Minor comments:

1) Part of Fig 1B is missing. Please provide the complete structure.

2) Key words should be arranged alphabetically.

Author Response

1) The abstract need to be more organized. The following line should be rearranged:

“The data also suggest that enhanced forms of arrestin do not bind GPCRs exactly like the wild type protein, calling for caution in interpretation of structures of the arrestin-receptor complex, in all of which different enhanced arrestin mutants and reengineered receptors were used.”

Thanks! This sentence was changed: it was broken down in two and reworded.

2) On pg.11; paragraph 2, the authors mention that the mutations of residues Pro 134 and Pro136 increase the binding; this needs to be explained with more clarity. Some modeling experiments with ligand-receptor docking and scoring are necessary. The molecular interactions between arrestin and receptor will improve the quality of this paper.

Thanks! Indicated part of the manuscript was modified. The most likely explanation of the data was proposed.

We respectfully disagree with the suggestion of adding modeling. In fact, the predictions of modeling must be tested by experiments, not the other way around.  

3) The binding energy values of the WT arrestin - receptor and mutant arrestin- receptor complexes should be provided.

For technical reasons, the arrestin-receptor binding energy cannot be measured. Arrhenius activation energy for WT arrestin-1 binding to P-Rh* was estimated at 165 kJ/mole (Biochemistry 28 (4), 1770-5 (1989)). Temperature dependence of the binding suggests that in case of enhanced mutants this energy barrier is significantly lower (Prog Retin Eye Res 30 (6), 405-30 (2011)).

4) The uniqueness of the experimental methods used in this study needs to be explained with more clarity.

Thanks! We explained the method more clearly at the beginning of the Results section.

5) What are the applications of this work as for example the author briefly mention in the discussion section how some of these mutations can be used to design therapeutic protein structures.

Thanks! We explicitly explained potential usefulness of phosphorylation-independent arrestin mutants with high binding to unphosphorylated GPCRs for gene therapy of congenital disorders associated with defects in receptor phosphorylation.

6) Too many self citations.

We limited citations to relevant papers regardless of authorship. Direct binding assay that we developed in 1992 (J Biol Chem 267 (30), 21919-23 (1992); not cited to minimize self-citations) allows to functionally characterize numerous arrestin mutants without requiring expression and purification of all these proteins. That saves a lot of time and effort. We took advantage of this method, which yielded insights into the molecular mechanisms involved in the interaction long before the first structure of arrestin-1 was solved, for the last 30 years. The data obtained using this method explained how arrestins detect phosphates attached to hundreds of GPCRs with diverse sequences of phosphorylated intracellular parts, revealed critical intramolecular interactions stabilizing the basal arrestin conformation that are broken by bound GPCRs (which enabled the construction of enhanced arrestins), etc. These conclusions were confirmed years later by the structures of free and receptor-bound arrestins. That is the reason why many of the relevant papers describe our work.   

Minor comments:

  • Part of Fig 1B is missing. Please provide the complete structure.

In Fig. 2B we focused on the arrestin-rhodopsin interface under study. We considered the parts of rhodopsin that do not contact arrestin-1 in the complex not directly relevant to the work presented. The full structure of the arrestin-1 complex with rhodopsin was published (Nature 523 (7562), 561-7 (2015); Cell 170 (3), 457-469 (2017)), deposited in the structure database (4ZWJ), and is readily available.

2) Key words should be arranged alphabetically.

Thanks! Done.